# POMDPs in Continuous Time and Discrete Spaces

**Bastian Alt**[*]     **Matthias Schultheis**[*,‡]     **Heinz Koeppl**[*,‡]
[*] Department of Electrical Engineering and Information Technology
[‡] Centre for Cognitive Science
Technische Universität Darmstadt
{bastian.alt, matthias.schultheis, heinz.koeppl}@bcs.tu-darmstadt.de

## Abstract

Many processes, such as discrete event systems in engineering or population dynamics in biology, evolve in discrete space and continuous time. We consider the problem of optimal decision making in such discrete state and action space systems under partial observability. This places our work at the intersection of optimal filtering and optimal control. At the current state of research, a mathematical description for simultaneous decision making and filtering in continuous time with finite state and action spaces is still missing. In this paper, we give a mathematical description of a continuous-time partial observable Markov decision process (POMDP). By leveraging optimal filtering theory we derive a Hamilton-Jacobi-Bellman (HJB) type equation that characterizes the optimal solution. Using techniques from deep learning we approximately solve the resulting partial integro-differential equation. We present (i) an approach solving the decision problem offline by learning an approximation of the value function and (ii) an online algorithm which provides a solution in belief space using deep reinforcement learning. We show the applicability on a set of toy examples which pave the way for future methods providing solutions for high dimensional problems.

## 1   Introduction

Continuous-time models have extensively been studied in machine learning and control. They are especially beneficial to reason about latent variables at time points which are not included in the data. In a broad range of topics such as natural language processing [49], social media dynamics [31] or biology [18] to name just a few, the underlying process naturally evolves continuously in time. In many applications the control of such time-continuous models is of interest. There exist already numerous approaches which tackle the control problem of continuous state space systems, however, for many processes a discrete state space formulation is more suited. This class of systems is discussed in the area of discrete event systems [10]. Decision making in these systems has a long history, yet, if the state is not fully observed acting optimally in such systems is notoriously hard. Many approaches resort to heuristics such as applying a separation principle between inference and control. Unfortunately, this can lead to weak performance as the agent does not incorporate effects of its decisions for future inference.

In the past, this problem was also approached by using a discrete time formulation such as a POMDP model [22]. Nevertheless, it is not always straight-forward to discretize the problem as it requires adding pseudo observations for time points without observations. Additionally, the time discretization can lead to problems when learning optimal controllers in the continuous-time setting [44].

A more principled way to approach this problem is to define the model in continuous time with a proper observation model and to solve the resulting model formulation. Still, it is not clear a priori, how to design such a model and even less how to control it in an optimal way. In this paper, we provide a formulation of this problem by introducing a continuous-time analogue to the POMDP

framework. We additionally show how methods from deep learning can be used to find approximate solutions for control under the continuous-time assumption. Our work can be seen as providing a first step towards scalable control laws for high dimensional problems by making use of further approximation methods. An implementation of our proposed method is publicly available[1].

**Related Work**

Historically, optimal control in continuous time and space is a classical problem and has been addressed ever since the early works of Pontryagin [35] and Kalman [23]. Continuous-time reinforcement learning formulations have been studied [13, 5, 48] and resulted in works such as the advantage updating and advantage learning algorithms [3, 17] and more recently in function approximation methods [44]. Continuous-time formulations in the case of full observability and discrete spaces are regarded in the context of semi-Markov decision processes (SMDPs) [4, 7], with applications e.g., in E-commerce [53] or as part of the options framework [43, 21].

An important area for the application of time-continuous discrete-state space models is discussed in the queueing theory literature [6]. Here, the state space describes the dynamics of the discrete number of customers in the queue. These models are used for example for internet communication protocols, such as in TCP congestion control. More generally, the topic is also considered within the area of stochastic hybrid systems [11], where mixtures of continuous and discrete state spaces are discussed. Though, the control under partial observability is often only considered under very specific assumptions on the dynamics.

A classical example for simultaneous optimal filtering and control in continuous space is the linear quadratic Gaussian (LQG) controller [41]. In case of partial observability and discrete spaces, the widely used POMDP framework [22] builds a sound foundation for optimal decision making in discrete time and is the basis of many modern methods, e.g., [37, 54, 19]. By applying discretizations, it was also used to solve continuous state space problems as discussed in [12]. Another existing framework which is close to our work is the partial observable semi-Markov decision process (POSMDP) [30] which has applications in fields such as robotics [34] or maintenance [40]. One major drawback of this framework is that observations are only allowed to occur at time points where the latent continuous-time Markov chain (CTMC) jumps. This assumption is actually very limiting, as in many applications observations are received at arbitrary time points or knowledge about jump times is not available.

The development of numerical solution methods for high dimensional partial differential equations (PDEs), such as the HJB equation, is an ongoing topic of research. Popular approaches include techniques based on linearization such as differential dynamic programming [20, 46], stochastic simulation techniques as in the path integral formalism [24, 47] and collocation-based approaches as in [38]. Latter have been extensively discussed due to the recent advances of function approximation by neural networks which have achieved huge empirical success [26]. Approaches solving HJB equations among other PDEs using deep learning can be found in [45, 29, 15, 39].

In the following section we introduce several results from optimal filtering and optimal control, which help to bring our work into perspective.

## 2   Background

**Continuous-Time Markov Chains.**   First, we discuss the basic notion of the continuous-time Markov chain (CTMC) $\{X(t)\}_{t \geq 0}$, which is a Markov model on discrete state space $\mathcal{X} \subseteq \mathbb{N}$ and in continuous time $t \in \mathbb{R}_{\geq 0}$. Its evolution for a small time step $h$ is characterized by $\mathbb{P}(X(t + h) = x \mid X(t) = x') = \mathbb{1}(x = x') + \Lambda(x', x)h + o(h)$, with $\lim_{h \to 0} \frac{o(h)}{h} = 0$. Here, $\Lambda : \mathcal{X} \times \mathcal{X} \to \mathbb{R}$ is the rate function, where $\Lambda(x', x)$ denotes the rate of switching from state $x'$ to state $x$. We define the exit rate of a state $x \in \mathcal{X}$ as $\Lambda(x) := \sum_{x' \neq x} \Lambda(x, x')$ and $\Lambda(x, x) := -\Lambda(x)$. Due to the memoryless property of the Markov process, the waiting times between two jumps of the chain is exponentially distributed. The definition can also be extended to time dependent rate functions, which correspond to a time inhomegenouos CTMC, for further details see [33].

**Optimal Filtering of Latent CTMCs.** In the case of partial observability, we do not observe the CTMC $X(t)$ directly, but observe a stochastic process $Y(t)$ depending on $X(t)$. In this setting the goal of optimal filtering [2] is to calculate the filtering distribution, which is also referred to as the belief state at time $t$,

$$\pi(x,t) := \mathbb{P}(X(t) = x \mid y_{[0,t)}),$$

where $y_{[0,t)} := \{y(s)\}_{s \in [0,t)}$ is the set of observations in the time interval $[0,t)$. The filtering distribution provides a sufficient statistic to the latent state $X(t)$. We denote the filtering distribution in vector form with components $\{\pi(x,t)\}_{x \in \mathcal{X}}$ by $\boldsymbol{\pi}(t) \in \Delta^{|\mathcal{X}|}$ and use $\Delta^{|\mathcal{X}|}$ for the continuous belief space which is a $|\mathcal{X}|$ dimensional probability simplex.

In general, a large class of filter dynamics follow the description of a jump diffusion process, see [16] for an introduction and the control thereof. The filter dynamics are dependent on an observation model, of which several have been discussed in the literature. A continuous-discrete filter [18] for example uses a model with covariates $\{y_i\}_{i=1,\dots,n}$ observed at discrete time points $t_1 < t_2 < \cdots < t_n$, i.e., $y_i := y(t_i)$, according to $y_i \sim p(y_i \mid x)$, where $p(y_i \mid x) := p(y_i \mid X(t_i) = x)$ is a probability density or mass function. In this setting the filtering distribution $\pi(x,t) = \mathbb{P}(X(t) = x \mid y_1, \dots, y_n)$, with $t_n < t$ follows the usual forward equation

$$\frac{\mathrm{d}\pi(x,t)}{\mathrm{d}t} = \sum_{x'} \Lambda(x',x)\pi(x',t), \tag{1}$$

between observation times $t_i$ and obeys the following reset conditions

$$\pi(x,t_i) = \frac{p(y_i \mid x)\pi(x,t_i^-)}{\sum_{x'} p(y_i \mid x')\pi(x',t_i^-)}, \tag{2}$$

at observation times where $\pi(x,t_i^-)$ denotes the belief just before the observation. The reset conditions at the observation time points represent the posterior distribution of $X(t)$ which combines the likelihood model $p(y \mid x)$ with the previous filtering distribution $\pi(x',t_i^-)$ as prior. The filter equations Eqs. (1) and (2) can be written in the differential form as

$$\mathrm{d}\pi(x,t) = \sum_{x'} \Lambda(x',x)\pi(x',t)\mathrm{d}t + [\pi(x,t_{N(t)}) - \pi(x,t)]\mathrm{d}N(t),$$

which is a special case of a jump diffusion without a diffusion part. The observations enter the filtering equation by the counting process $N(t) = \sum_{i=1}^{n} \mathbb{1}(t_i \leq t)$ and the jump amplitude $\pi(x,t_{N(t)}) - \pi(x,t)$, which sets the filter to the new posterior distribution $\pi(x,t_{N(t)})$, see Eq. (2).

Other optimal filters can be derived by solving the corresponding Kushner-Stratonovic equation [25, 36]. One instance is the Wonham filter [52, 18] which is discussed in Appendix A.1.

An important observation is that for the case of finite state spaces the filtering distribution is described by a finite dimensional object, opposed to the case of, e.g., uncountable state spaces, where the filtering distribution is described by a probability density, which is infinite dimensional. This is helpful as the filtering distribution can be used as a state in a control setting.

**Optimal Control and Reinforcement Learning.** Next, we review some results from continuous-time optimal control [41] and reinforcement learning [42], which can be applied to continuous state spaces. Consider a stochastic differential equation (SDE) of the form $\mathrm{d}\mathbf{X}(t) = \mathbf{f}(\mathbf{X}(t), u(t))\mathrm{d}t + \mathbf{G}(\mathbf{X}(t), u(t))\mathrm{d}\mathbf{W}(t)$, where dynamical evolution of the state $\mathbf{X}(t)$ is corrupted by noise following standard Brownian motion $\mathbf{W}(t)$. The control is a function $u : \mathbb{R}_{\geq 0} \to \mathcal{U}$, where $\mathcal{U}$ is an appropriate action space. In traditional stochastic optimal control the optimal value function, also named cost to go, of a controlled dynamical system is defined by the expected cumulative reward under the optimal policy,

$$V^*(\mathbf{x}) := \max_{u_{[t,\infty)}} \mathsf{E}\left[\int_t^\infty \frac{1}{\tau}e^{-\frac{s-t}{\tau}}R(\mathbf{X}(s), u(s))\,\mathrm{d}s \;\middle|\; \mathbf{X}(t) = \mathbf{x}\right], \tag{3}$$

where maximization is carried out over all control trajectories $u_{[t,\infty)} := \{u(s)\}_{s \in [t,\infty)}$. The performance measure or reward function is given by $R : \mathcal{X} \times \mathcal{U} \to \mathbb{R}$ and $\tau$ denotes the discount factor[2].

We use a normalization by $1/\tau$ for the value function as it was found to stabilize its learning process when function approximations are used [45]. The optimal value function is given by a PDE, the stochastic HJB equation

$$V^*(\mathbf{x}) = \max_{u \in \mathcal{U}} \left\{ R(\mathbf{x}, u) + \tau \frac{\partial V^*(\mathbf{x})}{\partial \mathbf{x}}^{\top} \mathbf{f}(\mathbf{x}, u) + \frac{\tau}{2} \operatorname{tr} \left( \frac{\partial^2 V^*(\mathbf{x})}{\partial \mathbf{x}^2} \mathbf{G}(\mathbf{x}, u) \mathbf{G}(\mathbf{x}, u)^{\top} \right) \right\}.$$

An optimal policy $\mu^* : \mathcal{X} \to \mathcal{U}$ can be found by solving the PDE and maximizing the right hand side of the HJB equation for every $x \in \mathcal{X}$.

For a deterministic policy $\mu : \mathcal{X} \to \mathcal{U}$ we define its value as the expected cumulative reward

$$V^\mu(\mathbf{x}) := \mathsf{E} \left[ \int_t^\infty \frac{1}{\tau} e^{-\frac{s-t}{\tau}} R(\mathbf{X}(s), U(s)) \, \mathrm{d}s \ \middle| \ \mathbf{X}(t) = \mathbf{x} \right], \tag{4}$$

where the control $U(s) = \mu(\mathbf{X}(s)), s \in [t, \infty)$ follows the policy $\mu$. If the state dynamics are deterministic and the state is differentiable in time, which is the case if there is no diffusion, i.e., $\mathbf{G}(\mathbf{x}, u) = \mathbf{0}$, one can derive a differential equation for the value function [13]. This is achieved by evaluating the value function $V^\mu(\mathbf{x}(t))$ at the function $\mathbf{x}(t)$ and then differentiating both sides of Eq. (4) by time resulting in

$$\tau \frac{\mathrm{d}}{\mathrm{d}t} V^\mu(\mathbf{x}(t)) = V^\mu(\mathbf{x}(t)) - R(\mathbf{x}(t), \mu(\mathbf{x}(t))). \tag{5}$$

The residuum of Eq. (5) can be identified as a continuous-time analog of the temporal difference (TD)-error

$$\delta(t) := r(t) - V^\mu(\mathbf{x}(t)) + \tau \frac{\mathrm{d}}{\mathrm{d}t} V^\mu(\mathbf{x}(t)),$$

which an agent should minimize using an observed reward signal $r(t) = R(\mathbf{x}(t), \mu(\mathbf{x}(t)))$ to solve the optimal control problem using reinforcement learning in an online fashion. In this presented case of stochastic optimal control we have assumed that the state is directly usable for the controller. In the partially observable setting, in contrast, this is not the case and the controller can only rely on observations of the state. Hence, the sufficient statistic, the belief state, has to be used for calculating the optimal action. Next, we describe how to model the discrete state space version of the latter case.

## 3 The Continuous-Time POMDP Model

In this paper we consider a continuous-time equivalent of a POMDP model with time index $t \in \mathbb{R}_{\geq 0}$ defined by the tuple $\langle \mathcal{X}, \mathcal{U}, \mathcal{Y}, \Lambda, \mathbb{P}_{Y|X,u}, R, \tau \rangle$.

We assume a finite state space $\mathcal{X}$ and a finite action space $\mathcal{U}$. The observation space $\mathcal{Y}$ is either a discrete space or an uncountable space. The latent controlled Markov process follows a CTMC with rate function $\Lambda : \mathcal{X} \times \mathcal{X} \times \mathcal{U} \to \mathbb{R}$, i.e., $\mathbb{P}(X(t+h) = x \mid X(t) = x', u(t) = u) = \mathbb{1}(x = x') + \Lambda(x', x \mid u)h + o(h)$, with exit rate $\Lambda(x \mid u) = \sum_{x' \neq x} \Lambda(x, x' \mid u)$. Note that the rate function $\Lambda$ is a function of the control variable. Therefore, the system dynamics are described by a time inhomogeneous CTMC. The underlying process $X(t)$ cannot be directly observed but an observation process $Y(t)$ is available providing information about $X(t)$. The observation model is specified by the conditional probability measure $\mathbb{P}_{Y|X,u}$. The reward function is given by $R : \mathcal{X} \times \mathcal{U} \to \mathbb{R}$. Throughout, we consider an infinite horizon problem with discount $\tau$. We denote the filtering distribution for the latent state $X(t)$ at time $t$ by $\pi(x, t) = \mathbb{P}(X(t) = x \mid y_{[0,t)}, u_{[0,t)})$ with belief state $\boldsymbol{\pi}(t) \in \Delta^{|\mathcal{X}|}$. This filtering distribution is used as state of the partially observable system. Additionally, we define $\mu : \Delta^{|\mathcal{X}|} \to \mathcal{U}$ as a deterministic policy, which maps a belief state to an action. The performance of a policy $\mu$ with control $U(s) = \mu(\boldsymbol{\pi}(s)), s \in [0, \infty)$ in the infinite horizon case is given analogously to the standard optimal control case by

$$J^\mu(\boldsymbol{\pi}) = \mathsf{E} \left[ \int_0^\infty \frac{1}{\tau} e^{-\frac{s}{\tau}} R(X(s), U(s)) \mathrm{d}s \ \middle| \ \boldsymbol{\pi}(0) = \boldsymbol{\pi} \right],$$

where the expectation is carried out w.r.t. both the latent state and observation processes.

## 3.1 Simulating the Model

First, we describe a generative process to draw trajectories from a continuous-time POMDP. Consider the stochastic simulation with given policy $\mu$ for a trajectory starting at time $t$ with state $X(t) = x'$, and belief $\boldsymbol{\pi}(t)$. The first step is to draw a waiting time $\xi$ of the latent CTMC after which the state switches. This CTMC is time inhomogeneous with exit rate $\Lambda(x' \mid \mu(\boldsymbol{\pi}(t)))$ since the control modulates the transition function. Therefore, we sample $\xi$ from the time dependent exponential distribution with CDF

$$P(\xi) = 1 - \exp(\int_t^{t+\xi} \Lambda(x' \mid \mu(\boldsymbol{\pi}(s)))\mathrm{d}s)$$

for which one needs to solve the filtering trajectory $\{\boldsymbol{\pi}(s)\}_{s \in [t, t+\xi)}$ using a numeric (stochastic) differential equation solver beforehand. There are multiple ways to sample from time dependent exponential distributions. A convenient method to jointly calculate the filtering distribution and the latent state trajectory is provided by the thinning algorithm [27]. For an adaptation for the continuous-time POMDP see Appendix B.1. As second step, after having sampled the waiting time $\xi$, we can draw the next state $X(t + \xi)$ given $\xi$ from the categorical distribution

$$\mathbb{P}(X(t + \xi) = x \mid x', \xi) = \begin{cases} \frac{\Lambda(x', x \mid \mu(\boldsymbol{\pi}(t+\xi)))}{\Lambda(x' \mid \mu(\boldsymbol{\pi}(t+\xi)))} & \text{if } x' \neq x, \\ 0 & \text{otherwise.} \end{cases}$$

The described process is executed for the initial belief and state at $t = 0$ and repeated until the total simulation time has been reached.

## 3.2 The HJB Equation in Belief Space

Next, we derive an equation for the value function, which can be solved to obtain an optimal policy. The infinite horizon optimal value function is given by the expected discounted cumulative reward

$$V^*(\boldsymbol{\pi}) = \max_{u_{[t,\infty)}} \mathsf{E}\left[ \int_t^\infty \frac{1}{\tau} e^{-\frac{s-t}{\tau}} R(X(s), u(s))\mathrm{d}s \;\middle|\; \boldsymbol{\pi}(t) = \boldsymbol{\pi} \right]. \tag{6}$$

The value function depends on the belief state $\boldsymbol{\pi}$ which provides a sufficient statistic for the state. By splitting the integral into two terms from $t$ to $t + h$ and from $t + h$ to $\infty$, we have

$$V^*(\boldsymbol{\pi}) = \max_{u_{[t,\infty)}} \mathsf{E}\left[ \int_t^{t+h} \frac{1}{\tau} e^{-\frac{s-t}{\tau}} R(X(s), u(s))\mathrm{d}s + \int_{t+h}^\infty \frac{1}{\tau} e^{-\frac{s-t}{\tau}} R(X(s), u(s))\mathrm{d}s \;\middle|\; \boldsymbol{\pi}(t) = \boldsymbol{\pi} \right]$$

and by identifying the second integral as $e^{-\frac{h}{\tau}} V^*(\boldsymbol{\pi}(t + h))$, we find the stochastic principle of optimality as

$$V^*(\boldsymbol{\pi}) = \max_{u_{[t,t+h)}} \mathsf{E}\left[ \int_t^{t+h} \frac{1}{\tau} e^{-\frac{s-t}{\tau}} R(X(s), u(s))\mathrm{d}s + e^{-\frac{h}{\tau}} V^*(\boldsymbol{\pi}(t + h)) \;\middle|\; \boldsymbol{\pi}(t) = \boldsymbol{\pi} \right]. \tag{7}$$

Here, we consider the class of filtering distributions which follow a jump diffusion process

$$\mathrm{d}\boldsymbol{\pi}(t) = \mathbf{f}(\boldsymbol{\pi}(t), u(t))\mathrm{d}t + \mathbf{G}(\boldsymbol{\pi}(t), u(t))\mathrm{d}\mathbf{W}(t) + \mathbf{h}(\boldsymbol{\pi}(t), u(t))\mathrm{d}N(t), \tag{8}$$

where $\mathbf{f} : \Delta^{|\mathcal{X}|} \times \mathcal{U} \to \mathbb{R}^{|\mathcal{X}|}$ denotes the drift function, $\mathbf{G} : \Delta^{|\mathcal{X}|} \times \mathcal{U} \to \mathbb{R}^{|\mathcal{X}| \times m}$ the dispersion matrix, with $\mathbf{W}(t) \in \mathbb{R}^m$ being an $m$ dimensional standard Brownian motion and $\mathbf{h} : \Delta^{|\mathcal{X}|} \times \mathcal{U} \to \mathbb{R}^{|\mathcal{X}|}$ denotes the jump amplitude. We assume a Poisson counting process $N(t) \sim \mathcal{PP}(N(t) \mid \lambda)$ with rate $\lambda$ for the observation times $\{t_i\}_{i \in \mathbb{N}}$, which implies that $t_i - t_{i-1} \sim \mathrm{Exp}(t_i - t_{i-1} \mid \lambda)$. By applying Itô's formula for the jump diffusion processes in Eq. (8) to Eq. (7), dividing both sides by $h$ and taking the limit $h \to 0$ we find the PDE

$$V^*(\boldsymbol{\pi}) = \max_{u \in \mathcal{U}} \left\{ \mathsf{E}\left[R(x, u) \mid \boldsymbol{\pi}\right] + \tau \frac{\partial V^*(\boldsymbol{\pi})}{\partial \boldsymbol{\pi}} \mathbf{f}(\boldsymbol{\pi}, u) + \frac{\tau}{2} \mathrm{tr}\left( \frac{\partial^2 V^*(\boldsymbol{\pi})}{\partial \boldsymbol{\pi}^2} \mathbf{G}(\boldsymbol{\pi}, u)\mathbf{G}(\boldsymbol{\pi}, u)^\top \right) \right. \\ \left. + \tau \lambda \left( \mathsf{E}\left[V^*(\boldsymbol{\pi} + \mathbf{h}(\boldsymbol{\pi}, u)\right] - V^*(\boldsymbol{\pi})) \right) \right\}, \tag{9}$$

where $\mathsf{E}\left[R(x, u) \mid \boldsymbol{\pi}\right] = \sum_x R(x, u)\pi(x)$. For a detailed derivation see Appendix A.2. We will focus mainly on the case of a controlled continuous-discrete filter

$$\mathrm{d}\pi(x, t) = \sum_{x'} \Lambda(x', x \mid u(t))\pi(x', t)\mathrm{d}t + \left( \frac{p(y_{N(t)} \mid x, u(t))\pi(x, t)}{\sum_{x'} p(y_{N(t)} \mid x', u(t))\pi(x', t)} - \pi(x, t) \right) \mathrm{d}N(t).$$

For this filter, the resulting HJB equation is given by the partial integro-differential equation

$$V^*(\boldsymbol{\pi}) = \max_{u \in \mathcal{U}} \left\{ \sum_x R(x, u)\pi(x) + \tau \sum_{x,x'} \frac{\partial V^*(\boldsymbol{\pi})}{\partial \pi(x)} \Lambda(x', x \mid u)\pi(x') + \tau\lambda \left( \int \sum_x p(y \mid x, u) \right. \right.$$
$$\left. \left. \pi(x)V^*(\boldsymbol{\pi}^+)\mathrm{d}y - V^*(\boldsymbol{\pi})) \right\},$$

with $\pi^+(x) = \frac{p(y|x,u)\pi(x)}{\sum_{x'} p(y|x',u)\pi(x')}$. A derivation for the HJB equation for the continuous-discrete filter and for a controlled Wonham filter is provided in Appendix A.3.

To find the optimal policy $\mu^*$ corresponding to the optimal value function $V^*(\boldsymbol{\pi})$, it is useful to define the optimal advantage function coinciding with Eq. (9) as

$$A^*(\boldsymbol{\pi}, u) := \mathsf{E}\left[R(x, u) \mid \boldsymbol{\pi}\right] - V^*(\boldsymbol{\pi}) + \tau \frac{\partial V^*(\boldsymbol{\pi})}{\partial \boldsymbol{\pi}} \mathbf{f}(\boldsymbol{\pi}, u)$$
$$+ \frac{\tau}{2} \operatorname{tr} \left( \frac{\partial^2 V^*(\boldsymbol{\pi})}{\partial \boldsymbol{\pi}^2} \mathbf{G}(\boldsymbol{\pi}, u)\mathbf{G}(\boldsymbol{\pi}, u)^\top \right) + \tau\lambda \left(\mathsf{E}\left[V^*(\boldsymbol{\pi} + \mathbf{h}(\boldsymbol{\pi}, u))\right] - V^*(\boldsymbol{\pi})\right). \tag{10}$$

In order to satisfy Eq. (9), the consistency equation

$$\max_{u \in \mathcal{U}} A^*(\boldsymbol{\pi}, u) = 0 \tag{11}$$

is required. The optimal policy can then be obtained as $\mu^*(\boldsymbol{\pi}) = \arg\max_{u \in \mathcal{U}} A^*(\boldsymbol{\pi}, u)$.

## 4 Algorithms

For the calculation of an optimal policy we have to find the advantage function. As it dependents on the value function, we have to calculate both of them. Since the PDE for the value function in Eq. (9) does not have a closed form solution, we present next two methods to learn the functions approximately.

### 4.1 Solving the HJB Equation Using a Collocation Method

For solving the HJB equation (9) we first apply a collocation method [38, 45, 29] with a parameterized value function $V_\phi(\boldsymbol{\pi})$, which is modeled by means of a neural network. We define the residual without the maximum operator of the HJB equation under the parameterization as the advantage

$$A_\phi(\boldsymbol{\pi}, u) := \mathsf{E}\left[R(x, u) \mid \boldsymbol{\pi}\right] - V_\phi(\boldsymbol{\pi}) + \tau \frac{\partial V_\phi(\boldsymbol{\pi})}{\partial \boldsymbol{\pi}} \mathbf{f}(\boldsymbol{\pi}, u)$$
$$+ \frac{\tau}{2} \operatorname{tr} \left( \frac{\partial^2 V_\phi(\boldsymbol{\pi})}{\partial \boldsymbol{\pi}^2} \mathbf{G}(\boldsymbol{\pi}, u)\mathbf{G}(\boldsymbol{\pi}, u)^\top \right) + \tau\lambda \left(\mathsf{E}\left[V_\phi(\boldsymbol{\pi} + \mathbf{h}(\boldsymbol{\pi}, u))\right] - V_\phi(\boldsymbol{\pi})\right).$$

For learning, we sample beliefs $\{\boldsymbol{\pi}^{(i)}\}_{i=1,\dots,N}$, with $\boldsymbol{\pi}^{(i)} \in \Delta^{|\mathcal{X}|}$, from some base distribution $\boldsymbol{\pi}^{(i)} \sim p(\boldsymbol{\pi})$ and estimate the optimal parameters $\hat\phi = \arg\min_\phi \sum_{i=1}^N \left\{\max_{u \in \mathcal{U}} A_\phi(\boldsymbol{\pi}^{(i)}, u)\right\}^2$ by minimizing the squared loss. If needed, we can approximate the expectation $\mathsf{E}\left[V_\phi(\boldsymbol{\pi}^{(i)} + \mathbf{h}(\boldsymbol{\pi}^{(i)}, u))\right]$ over the observation space by sampling. An algorithm for this procedure is given in Appendix B.2. For learning it is required to calculate the gradient $\frac{\partial V_\phi(\boldsymbol{\pi})}{\partial \boldsymbol{\pi}}$ and Hessian $\frac{\partial^2 V_\phi(\boldsymbol{\pi})}{\partial \boldsymbol{\pi}^2}$ of the value function w.r.t. the input $\boldsymbol{\pi}$. Generally, this can be achieved by automatic differentiation, but for a fully connected multi-layer feedforward network, the analytic expressions are given in Appendix A.4. The analytic expression makes it possible to calculate the gradient, Hessian and value function in one single forward pass [28].

Given the learned value function $V_{\hat\phi}$, we learn an approximation of the advantage function $A_\psi(\boldsymbol{\pi}, u)$ to obtain an approximately-optimal policy. To this end we use a parametrized advantage function $A_\psi(\boldsymbol{\pi}, u)$ and employ a collocation method to solve Eq. (10). To ensure the consistency Eq. (11) during learning, we apply a reparametrization [44, 50] as $A_\psi(\boldsymbol{\pi}, u) = \bar{A}_\psi(\boldsymbol{\pi}, u) - \max_{u' \in \mathcal{U}} \bar{A}_\psi(\boldsymbol{\pi}, u')$. The optimal parameters are found by minimizing the squared loss as

$$\hat\psi = \arg\min_\psi \sum_{i=1}^N \sum_{u \in \mathcal{U}} \left\{ \bar{A}_\psi(\boldsymbol{\pi}^{(i)}, u) - \max_{u'} \bar{A}_\psi(\boldsymbol{\pi}^{(i)}, u') - A_{\hat\phi}(\boldsymbol{\pi}^{(i)}, u) \right\}^2.$$

The corresponding policy can then be easily determined as $\mu(\boldsymbol{\pi}) = \arg\max_{u \in \mathcal{U}} A_{\hat{\psi}}(\boldsymbol{\pi}, u)$ using a single forward pass through the learned advantage function.

## 4.2 Advantage Updating

The HJB equation (9) can also be solved online using reinforcement learning techniques. We apply the advantage updating algorithm [3] and solve Eq. (10) by employing neural network function approximators for both the value function $V_\phi(\boldsymbol{\pi})$ and the advantage function $A_\psi(\boldsymbol{\pi}, u)$. The expectations in Eq. (10) are estimated using sample runs of the POMDP. Hence, the residual error can be calculated as

$$E_{\phi,\psi}(t) = A_\psi(\boldsymbol{\pi}(t), u(t)) - r(t) + V_\phi(\boldsymbol{\pi}(t)) - \tau \frac{\partial V_\phi(\boldsymbol{\pi}(t))}{\partial \boldsymbol{\pi}} \mathbf{f}(\boldsymbol{\pi}(t), u(t))$$
$$- \frac{\tau}{2} \operatorname{tr}\left( \frac{\partial^2 V_\phi(\boldsymbol{\pi}(t))}{\partial \boldsymbol{\pi}^2} \mathbf{G}(\boldsymbol{\pi}(t), u(t)) \mathbf{G}(\boldsymbol{\pi}(t), u(t))^\top \right) - \tau \lambda \left( V_\phi(\boldsymbol{\pi}(t^+)) - V_\phi(\boldsymbol{\pi}(t)) \right),$$

which can also be seen in terms of a continuous-time TD-error $\delta_\phi(t)$ as $E_{\phi,\psi}(t) = A_\psi(\boldsymbol{\pi}(t), u(t)) - \delta_\phi(t)$. Again we apply the reparametrization $A_\psi(\boldsymbol{\pi}, u) = \bar{A}_\psi(\boldsymbol{\pi}, u) - \max_{u' \in \mathcal{U}} \bar{A}_\psi(\boldsymbol{\pi}, u')$ to satisfy Eq. (11). For estimating the optimal parameters $\hat{\phi}$ and $\hat{\psi}$ we minimize the sum of squared residual errors.

The data for learning is generated by simulating episodes under an exploration policy as in [44]. As exploration policy, we employ a time variant policy $\tilde{\mu}(\boldsymbol{\pi}, t) = \arg\max_{u \in \mathcal{U}} \{ A_\psi(\boldsymbol{\pi}, u) + \epsilon(u, t) \}$, where $\epsilon(u, t)$ is a stochastic exploration process which we choose as the Ornstein-Uhlenbeck process $\mathrm{d}\epsilon(u, t) = -\kappa \epsilon(u, t) \mathrm{d}t + \sigma \mathrm{d}W(t)$. Generated trajectories are subsampled and saved to a replay buffer [32] which is used to provide data for the training procedure.

## 5 Experiments

**Experimental Tasks.** We tested our derived methods on several toy tasks of continuous-time POMDPs with discrete state and action space: An adaption of the popular tiger problem [9], a decentralized multi-agent network transmission problem [8] implementing the slotted aloha protocol, and a grid world problem. All problems are adapted to continuous time and observations at discrete time points. In the following, we provide a brief overview over the considered problems. A more detailed description containing the defined spaces and parameters can be found in Appendix C.

In the tiger problem, the state consists of the position of a tiger (*left/right*) and the agent has to decide between three actions for either improving his belief (*listen*) or exploiting his knowledge to avoid the tiger (*left/right*). While listening the agent can wait for an observation.

In the transmission problem, a set of stations has to adjust their sending rate in order to successfully transmit packages over a single channel as in the slotted Aloha protocol. Each station might have a package to send or not, and the only observation is the past state of the channel, which can be either *idle*, *transmission*, or *collision*. New packages arrive with a fixed rate at the stations. As for each number of available packages – with the exception of no packages available – there is a unique optimal action, a finite number of transmission rates as actions is sufficient.

In the grid world problem, an agent has to navigate through a grid world by choosing the directions in which to move next. The agent transitions with an exponentially distributed amount of time and, while doing so, can slip with some probability so that he instead moves into another direction. The agent receives only noisy information of his position from time to time.

**Results.** Both the offline collocation method and the online advantage updating method manage to learn reasonable approximations of the value and advantage functions for the considered problems.

For the tiger problem, the learned value and advantage function over the whole belief space are visualized in Fig. 1. The parabolic shape of the value function correctly indicates that higher certainty of the belief about the tiger's position results in a higher expected discounted cumulative reward as the agent can exploit this knowledge to omit the tiger. Note that the value function learned by the online advantage updating method differs marginally from the one learned by collocation in shape. This is due to the fact that in the advantage updating method only actually visited belief states are used in the learning process to approximate the value function and points in between need to be interpolated.

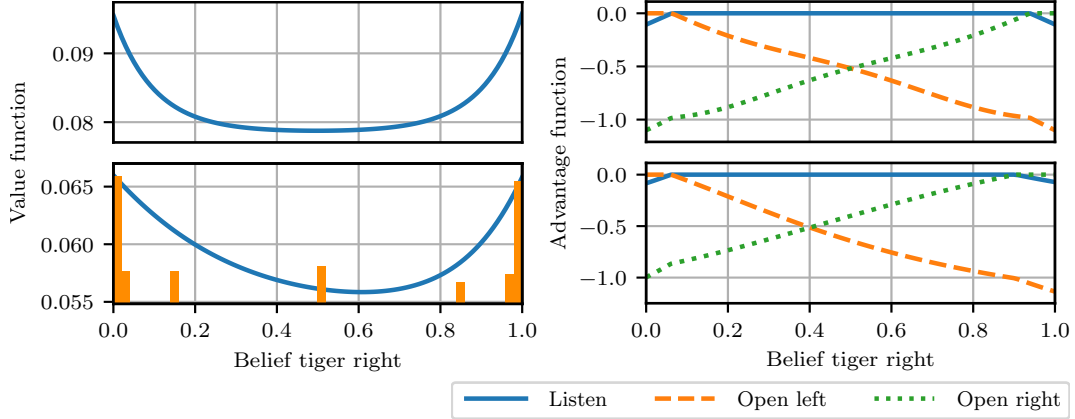

Figure 1: Learned value and advantage function for the continuous-time tiger problem. The upper plots show the approximated functions learned by the offline collocation method while for lower plots, the online advantage updating method was applied. The orange bars in the lower left plot show the proportions of the beliefs encountered during online learning. The advantage functions on the right can be used to determine the policy of the optimal agent.

The advantage function correctly indicates that for uncertain beliefs it is advantageous to first gain more information by executing the action *listen*. On the other hand, for certain beliefs directly opening the respective door is more useful in terms of reward and opening the opposed door considered as even more disadvantageous in these cases.

Results for the gridworld problem are visualized in Fig. 2 which shows the learned value and advantage function using the online advantage updating method. The figure visualizes the resulting values for certain beliefs, i.e., being at the respective fields with probability one. As expected, the learned value function assigns higher values to fields which are closer to the goal position $(3, 2)$. Actions leading to these states have higher advantage values. For assessing results for uncertain beliefs which are actually encountered when running the system, the figure also contains a sample run which successfully directs the agent to the goal. Results for the collocation method are found to be very similar. A respective plot can be found in Appendix C.2.

For the slotted Aloha transmission problem, a random execution of the policy learned by the offline collocation method is shown in Fig. 3. The upper two plots show the true states, i.e., number of packages and past system state, and the agent's beliefs which follow the prior dynamics of the system and jump when new observations are made. In the plot at the bottom, the learned advantage function and the resulting policy for the encountered beliefs are visualized. As derived in the appendix, in case of perfect information of the number of packages $n$, it is optimal to execute action $n - 1$ with exception of $n = 0$ where the action does not matter. When facing uncertainty, however, an optimistic behavior which is also reflected by the learned policy is more reasonable: As for a lower number of packages, the probability of sending a package and therefore

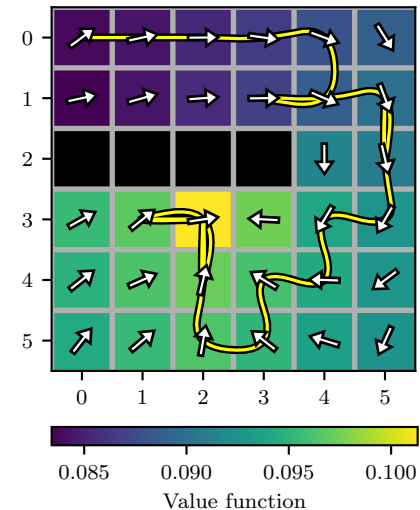

Figure 2: Learned value and advantage function for certain beliefs in the continuous-time gridworld problem using the advantage updating method. Being at the goal at position $(3, 2)$ results in a reward for the agent. The black fields in row 3 represent a wall that cannot be crossed. Colors of the fields indicate the approximated value function while arrows show the proportions of the advantage functions among different actions. The yellow curvy path indicates a respective random run starting at field (0,0) under the resulting policy.

collecting higher reward is higher. Thus, in case of uncertainty, one should opt for a lower action than the one executed under perfect information. Results for the online advantage updating method

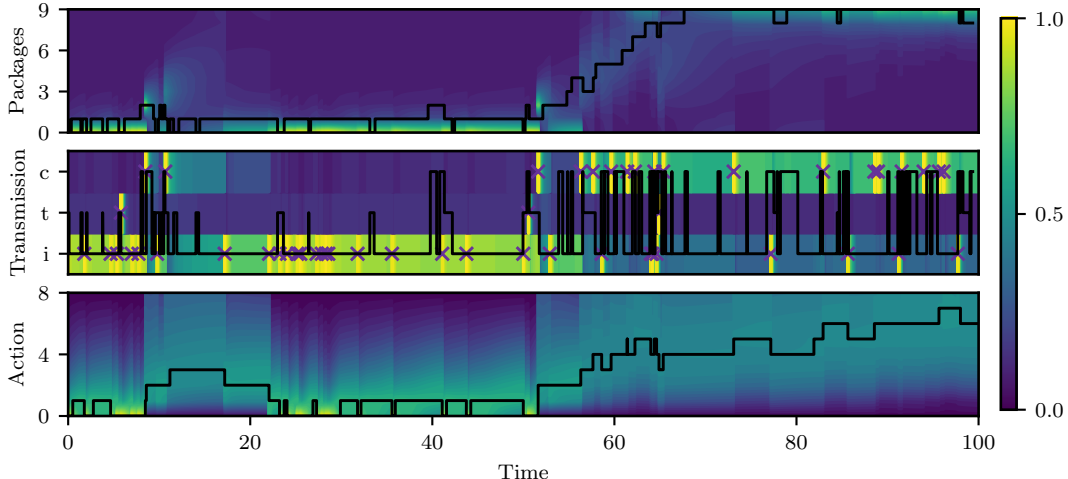

Figure 3: A sample run for the slotted Aloha transmission problem using a policy learned by the collocation method. The upper plot shows the actual number of packages available in the system while the middle one shows the past system state which can be either *idle* (i), *transmission* (t), or *collision* (c). The background of both plots indicate the marginal belief of number of packages and past system state, respectively. The past system states that are observed at discrete time points are indicated by a cross. The lower plot shows the policy of the agent resulting from the learned advantage function while the per timestep normalized advantage function is indicated by the background.

look qualitatively equal reflecting the same reasonable behavior. A respective plot is omitted here but can be found in Appendix C.2.

## 6 Conclusion

In this work we introduced a new model for decision making in continuous time under partial observability. We presented (i) a collocation-based offline method and (ii) an advantage updating online algorithm, which both find approximate solutions by the use of deep learning techniques. For evaluation we discussed qualitatively the found solutions for a set of toy problems that were adapted from literature. The solutions have shown to represent reasonable optimal decisions for the given problems.

In the future we are interested in exploring ways for making the proposed model applicable to more realistic large-scale problems. First, throughout this work we made the assumption of a known model. In many applications, however, this might not be the case and investigating how to realize dual control methods [14, 1] might be a fruitful direction. New scalable techniques for estimating parameters of latent CTMCs which could be used, are discussed in [51] but also learning the filtering distribution directly from data might be an option if the model is not available [19]. An issue we faced, was that for high dimensional problems the algorithms seemed to slowly converge to the optimal solution as the belief space grows linearly in the number of dimensions w.r.t. the number of states of the latent process. The introduction of variational and sampling methods seems to be promising to project the filtering distribution to a lower dimensional space and make solution of high-dimensional problems feasible. This could also enable the extension to continuous state spaces, where the filtering distribution is in general infinite dimensional and has to be projected to a finite dimensional representation, e.g., as it is done in assumed density filtering [36]. This will enable the use for interesting applications for example in queuing networks or more generally, in stochastic hybrid systems.

### Acknowledgments

This work has been funded by the German Research Foundation (DFG) as part of the project B4 within the Collaborative Research Center (CRC) 1053 – MAKI, and by the European Research Council (ERC) within the CONSYN project, grant agreement number 773196.

## Broader Impact

Not applicable to this manuscript.

## Footnotes

[1] https://git.rwth-aachen.de/bcs/pomdps_continuous_time_discrete_spaces

[2]Note that the discount factor is sometimes defined using certain transformations, e.g., $\frac{1}{\tau} = \rho = -\log\gamma$.

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
