[Supplementary Material]

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

## A    Models and Derivations

### A.1    The Wonham filter

Another well known observation model is

$$\mathrm{d}\mathbf{Y}(t) = \mathbf{g}(\mathbf{X}(t), t)\mathrm{d}t + \mathbf{B}(t)\mathrm{d}\mathbf{W}(t).$$

Here, the $l$-dimensional observation process $\mathbf{Y}(t) \in \mathbb{R}^l$ is described by an SDE with a drift function $\mathbf{g} : \mathcal{X} \times \mathbb{R}_{\geq 0} \to \mathbb{R}^l$ depending on the latent state of the CTMC. Noise is added by the dispersion matrix $\mathbf{B}(t) \in \mathbb{R}^{l \times m}$ and the Brownian motion $\mathbf{W}(t) \in \mathbb{R}^m$. The filtering distribution, here, can be described by the Wonham filter [52, 18]

$$\mathrm{d}\pi(x, t) = \sum_{x'} \Lambda(x', x)\pi(x', t)\mathrm{d}t + \pi(x, t)(\mathbf{g}(x, t) - \bar{\mathbf{g}}(t))^\top (\mathbf{B}(t)\mathbf{B}(t)^\top)^{-1}(\mathrm{d}\mathbf{Y}(t) - \bar{\mathbf{g}}(t)\mathrm{d}t),$$

where $\bar{\mathbf{g}}(t) = \sum_{x'} \mathbf{g}(x', t)\pi(x', t)$.

### A.2    The HJB Equation for a Filter Given by a Jump Diffusion Process

The principle of optimality reads

$$V^*(\boldsymbol{\pi}) = \max_{u_{[t,t+h)}} \mathsf{E}\left[\int_t^{t+h} \frac{1}{\tau}e^{-\frac{s-t}{\tau}}R(X(s), u(s))\mathrm{d}s + e^{-\frac{h}{\tau}}V^*(\boldsymbol{\pi}(t+h)) \,\bigg|\, \boldsymbol{\pi}(t) = \boldsymbol{\pi}\right]. \quad (12)$$

Assuming a jump diffusion dynamic as

$$\mathrm{d}\boldsymbol{\pi}(t) = \mathbf{f}(\boldsymbol{\pi}(t), u(t))\mathrm{d}t + \mathbf{G}(\boldsymbol{\pi}(t), u(t))\mathrm{d}\mathbf{W}(t) + \mathbf{h}(\boldsymbol{\pi}(t), u(t))\mathrm{d}N(t),$$

we apply Itô's formula for jump diffusion processes to the value function and find

$$
\begin{aligned}
V^*(\boldsymbol{\pi}(t+h)) = {}& V^*(\boldsymbol{\pi}(t)) + \int_t^{t+h} \frac{\partial V^*(\boldsymbol{\pi}(s))}{\partial \boldsymbol{\pi}}\mathbf{f}(\boldsymbol{\pi}(s), u(s))\mathrm{d}s \\
& + \int_t^{t+h} \frac{\partial V^*(\boldsymbol{\pi}(s))}{\partial \boldsymbol{\pi}}\mathbf{G}(\boldsymbol{\pi}(s), u(s))\mathrm{d}\mathbf{W}(s) \\
& + \int_t^{t+h} \frac{1}{2}\operatorname{tr}\left(\frac{\partial^2 V^*(\boldsymbol{\pi}(s))}{\partial \boldsymbol{\pi}^2}\mathbf{G}(\boldsymbol{\pi}(s), u(s))\mathbf{G}(\boldsymbol{\pi}(s), u(s))^\top\right)\mathrm{d}s \\
& + \int_t^{t+h} V^*(\boldsymbol{\pi}(s) + \mathbf{h}(\boldsymbol{\pi}(s)), u(s)) - V^*(\boldsymbol{\pi}(s))\mathrm{d}N(s).
\end{aligned}
\quad (13)
$$

By inserting Eq. (13) into Eq. (12) we find

$$
V^*(\boldsymbol{\pi}) = \max_{u_{[t,t+h)}} \mathsf{E} \left[ \int_t^{t+h} \frac{1}{\tau} e^{-\frac{s-t}{\tau}} R(X(s), u(s)) \mathrm{d}s + e^{-\frac{h}{\tau}} \left( V^*(\boldsymbol{\pi}(t)) \right. \right.
$$

$$
+ \int_t^{t+h} \frac{\partial V^*(\boldsymbol{\pi}(s))}{\partial \boldsymbol{\pi}} \mathbf{f}(\boldsymbol{\pi}(s), u(s)) \mathrm{d}s + \int_t^{t+h} \frac{\partial V^*(\boldsymbol{\pi}(s))}{\partial \boldsymbol{\pi}} \mathbf{G}(\boldsymbol{\pi}(s), u(s)) \mathrm{d}\mathbf{W}(s)
$$

$$
+ \int_t^{t+h} \frac{1}{2} \operatorname{tr} \left( \frac{\partial^2 V^*(\boldsymbol{\pi}(s))}{\partial \boldsymbol{\pi}^2} \mathbf{G}(\boldsymbol{\pi}(s), u(s)) \mathbf{G}(\boldsymbol{\pi}(s), u(s))^\top \right) \mathrm{d}s
$$

$$
\left. \left. + \int_t^{t+h} V^*(\boldsymbol{\pi}(s) + \mathbf{h}(\boldsymbol{\pi}(s)), u(s)) - V^*(\boldsymbol{\pi}(s)) \mathrm{d}N(s) \right) \,\middle|\, \boldsymbol{\pi}(t) = \boldsymbol{\pi} \right].
$$

Collecting terms in $V^*(\boldsymbol{\pi})$ and dividing both sides by $h$ we get

$$
V^*(\boldsymbol{\pi}) \frac{1 - e^{-\frac{h}{\tau}}}{h} = \max_{u_{[t,t+h)}} \mathsf{E} \left[ \frac{1}{h} \int_t^{t+h} \frac{1}{\tau} e^{-\frac{s-t}{\tau}} R(X(s), u(s)) \mathrm{d}s + \frac{e^{-\frac{h}{\tau}}}{h} \right(
$$

$$
\int_t^{t+h} \frac{\partial V^*(\boldsymbol{\pi}(s))}{\partial \boldsymbol{\pi}} \mathbf{f}(\boldsymbol{\pi}(s), u(s)) \mathrm{d}s + \int_t^{t+h} \frac{\partial V^*(\boldsymbol{\pi}(s))}{\partial \boldsymbol{\pi}} \mathbf{G}(\boldsymbol{\pi}(s), u(s)) \mathrm{d}\mathbf{W}(s)
$$

$$
+ \int_t^{t+h} \frac{1}{2} \operatorname{tr} \left( \frac{\partial^2 V^*(\boldsymbol{\pi}(s))}{\partial \boldsymbol{\pi}^2} \mathbf{G}(\boldsymbol{\pi}(s), u(s)) \mathbf{G}(\boldsymbol{\pi}(s), u(s))^\top \right) \mathrm{d}s
$$

$$
\left. + \int_t^{t+h} V^*(\boldsymbol{\pi}(s) + \mathbf{h}(\boldsymbol{\pi}(s)), u(s)) - V^*(\boldsymbol{\pi}(s)) \mathrm{d}N(s) \,\middle|\, \boldsymbol{\pi}(t) = \boldsymbol{\pi} \right].
$$

Taking $\lim_{h\to 0}$ and calculating the expectation w.r.t. $\mathbf{W}(t)$ and $N(t)$ we find

$$
\frac{1}{\tau} V^*(\boldsymbol{\pi}) = \max_u \frac{1}{\tau} \mathsf{E} \left[ R(X(t), u) \mid \boldsymbol{\pi}(t) = \boldsymbol{\pi} \right] + \frac{\partial V^*(\boldsymbol{\pi})}{\partial \boldsymbol{\pi}} \mathbf{f}(\boldsymbol{\pi}, u)
$$

$$
+ \frac{1}{2} \operatorname{tr} \left( \frac{\partial^2 V^*(\boldsymbol{\pi})}{\partial \boldsymbol{\pi}^2} \mathbf{G}(\boldsymbol{\pi}, u) \mathbf{G}(\boldsymbol{\pi}, u)^\top \right) + \lambda \left( \mathsf{E} \left[ V^*(\boldsymbol{\pi} + \mathbf{h}(\boldsymbol{\pi})) \right] - V^*(\boldsymbol{\pi}) \right),
$$

for further details, see [16].

### A.3 The HJB Equation for the Presented Filters

### A.3.1 The HJB Equation for the Continuous Discrete Filter

The HJB equation in belief space is

$$
V^*(\boldsymbol{\pi}) = \max_{u \in \mathcal{U}} \left\{ \mathsf{E} \left[ R(x, u) \mid \boldsymbol{\pi} \right] + \tau \frac{\partial V^*(\boldsymbol{\pi})}{\partial \boldsymbol{\pi}} \mathbf{f}(\boldsymbol{\pi}, u) + \frac{\tau}{2} \operatorname{tr} \left( \frac{\partial^2 V^*(\boldsymbol{\pi})}{\partial \boldsymbol{\pi}^2} \mathbf{G}(\boldsymbol{\pi}, u) \mathbf{G}(\boldsymbol{\pi}, u)^\top \right) \right.
$$

$$
\left. + \tau \lambda \left( \mathsf{E} \left[ V^*(\boldsymbol{\pi} + \mathbf{h}(\boldsymbol{\pi}, u)] - V^*(\boldsymbol{\pi}) \right) \right\}.
$$

The filter dynamics for the controlled continuous-discrete filter are given by

$$
\mathrm{d}\pi(x, t) = \sum_{x'} \Lambda(x', x \mid u(t)) \pi(x', t) \mathrm{d}t + \left( \frac{p(y_{N(t)} \mid x, u(t)) \pi(x, t)}{\sum_{x'} p(y_{N(t)} \mid x', u(t)) \pi(x', t)} - \pi(x, t) \right) \mathrm{d}N(t).
$$

Hence, the components $\{f(\boldsymbol{\pi}, u, x)\}_{x \in \mathcal{X}}$ of the drift function $f(\boldsymbol{\pi}, u)$ are given by

$$
f(\boldsymbol{\pi}, u, x) = \sum_{x'} \Lambda(x', x \mid u) \pi(x')
$$

and the diffusion term is zero, i.e., $\mathbf{G}(\boldsymbol{\pi}, u) = \mathbf{0}$. The components $\{h(\boldsymbol{\pi}, u, x)\}_{x \in \mathcal{X}}$ of the jump amplitude $\mathbf{h}(\boldsymbol{\pi}, u)$ are

$$
h(\boldsymbol{\pi}, u, x) = \frac{p(y \mid x, u) \pi(x)}{\sum_{x'} p(y \mid x', u) \pi(x')} - \pi(x),
$$

with $y \sim p(y)$ and $p(y) = \sum_x p(y \mid x, u)\pi(x)$. Thus, the expectation yields

$$\mathsf{E}\left[V^*(\boldsymbol{\pi} + \mathbf{h}(\boldsymbol{\pi}, u))\right] = \int \sum_x p(y \mid x, u)\pi(x)V^*(\boldsymbol{\pi}^+)\mathrm{d}y,$$

with $\pi^+(x) = \frac{p(y|x,u)\pi(x)}{\sum_{x'} p(y|x',u)\pi(x')}$. Finally, we have the HJB equation

$$V^*(\boldsymbol{\pi}) = \max_{u \in \mathcal{U}} \left\{ \sum_x R(x,u)\pi(x) + \tau \sum_{x,x'} \frac{\partial V^*(\boldsymbol{\pi})}{\partial \pi(x)} \Lambda(x', x \mid u)\pi(x') + \tau\lambda \left( \int \sum_x p(y \mid x, u) \right. \right.$$
$$\left. \left. \pi(x)V^*(\boldsymbol{\pi}^+)\mathrm{d}y - V^*(\boldsymbol{\pi}) \right) \right\}.$$

### A.3.2  The HJB Equation for the Wonham Filter

We consider the HJB equation in belief space

$$V^*(\boldsymbol{\pi}) = \max_{u \in \mathcal{U}} \left\{ \mathsf{E}\left[R(x,u) \mid \boldsymbol{\pi}\right] + \tau \frac{\partial V^*(\boldsymbol{\pi})}{\partial \boldsymbol{\pi}} \mathbf{f}(\boldsymbol{\pi}, u) + \frac{\tau}{2} \operatorname{tr}\left( \frac{\partial^2 V^*(\boldsymbol{\pi})}{\partial \boldsymbol{\pi}^2} \mathbf{G}(\boldsymbol{\pi}, u)\mathbf{G}(\boldsymbol{\pi}, u)^\top \right) \right.$$
$$\left. + \tau\lambda \left( \mathsf{E}\left[V^*(\boldsymbol{\pi} + \mathbf{h}(\boldsymbol{\pi}, u))\right] - V^*(\boldsymbol{\pi}) \right) \right\}$$

and the controlled Wonham filter

$$\mathrm{d}\pi(x, t) = \sum_{x'} \Lambda(x', x \mid u(t))\pi(x', t)\mathrm{d}t$$
$$+ \pi(x, t)(\mathbf{g}(x, u(t)) - \bar{\mathbf{g}}(u(t), t)(\mathbf{B}\mathbf{B}^\top)^{-1}(\mathrm{d}\mathbf{Y}(t) - \bar{\mathbf{g}}(u(t), t)\mathrm{d}t),$$

with $\bar{\mathbf{g}}(u(t), t) = \sum_{x'} \mathbf{g}(x', u(t))\pi(x', t)$. Therefore, we have the components $\{f(\boldsymbol{\pi}, u, x)\}_{x \in \mathcal{X}}$ of the drift function $\mathbf{f}(\boldsymbol{\pi}, u)$ as

$$f(\boldsymbol{\pi}, u, x) = \sum_{x'} \Lambda(x', x \mid u)\pi(x') + \pi(x)(\mathbf{g}(x, u) - \bar{\mathbf{g}}(u))^\top (\mathbf{B}\mathbf{B}^\top)^{-1}(\mathbf{g}(x, u) - \bar{\mathbf{g}}(u)).$$

The row-wise components $\{\mathbf{G}(\boldsymbol{\pi}, u, x)\}_{x \in \mathcal{X}}$ of the dispersion matrix $\mathbf{G}(\boldsymbol{\pi}, u)$ read

$$\mathbf{G}(\boldsymbol{\pi}, u, x) = \pi(x)\mathbf{B}^\top (\mathbf{B}\mathbf{B}^\top)^{-1}(\mathbf{g}(x, u) - \bar{\mathbf{g}}(u)),$$

with $\bar{\mathbf{g}}(u) = \sum_{x'} \mathbf{g}(x', u)\pi(x')$. The jump amplitude is zero, i.e., $\mathbf{h}(\boldsymbol{\pi}, u) = \mathbf{0}$.

Hence, we have the stochastic HJB equation

$$V^*(\boldsymbol{\pi}) = \max_u \sum_x R(x, u)\pi(x) + \frac{\partial V^*(\boldsymbol{\pi})}{\partial \boldsymbol{\pi}} \mathbf{f}(\boldsymbol{\pi}, u) + \frac{\tau}{2} \left( \frac{\partial^2 V^*(\boldsymbol{\pi})}{\partial \boldsymbol{\pi}^2} \mathbf{G}(\boldsymbol{\pi}, u)\mathbf{G}(\boldsymbol{\pi}, u)^\top \right).$$

### A.4  Gradient and Hessian of a Neural Network

In this section we derive the gradient and Hessian of the input of a value network using back-propagation.

Consider a $H$-Layer deep Neural network parametrizing a value function $V(\mathbf{x})$ as

$$V(\mathbf{x}) = \mathbf{W}^H \mathbf{z}^{H-1} + \vartheta^H$$
$$\mathbf{z}^h = \sigma(\mathbf{a}^h), \quad h = 0, \dots, H - 1$$
$$\mathbf{a}^h = \mathbf{W}^h \mathbf{z}^{h-1} + \vartheta^h, \quad h = 1, \dots, H - 1$$
$$\mathbf{a}^0 = \mathbf{W}^0 \mathbf{x} + \vartheta^0,$$

with input $\mathbf{x}$, weights $\mathbf{W}^h$, biases $\vartheta^h$ and activation function $\sigma$. Component wise, the equations are given by

$$z_k^h = \sigma(a_k^h), \quad h = 0, \dots, H - 1$$
$$a_k^h = \sum_n w_{kn}^h z_n^{h-1} + \vartheta_k^h, \quad h = 1, \dots, H - 1$$
$$a_k^0 = \sum_n w_{kn}^0 x_n + \vartheta_k^0.$$

### A.4.1 The Gradient Computation

Next we calculate the gradient w.r.t. the input i.e.,, $\partial_{\mathbf{x}} V(\mathbf{x})$. The component wise calculation yield

$$\partial_{x_i} V(\mathbf{x}) = \partial_{x_i} \sum_n w_n^H z_n^{H-1} + \vartheta^H$$

$$= \sum_n w_n^H \partial_{x_i} z_n^{H-1}.$$

where we compute

$$\partial_{x_i} z_k^h = \sigma'(a_k^h) \sum_n w_{kn}^h \partial_{x_i} z_n^{h-1}$$

and $\sigma'$ denotes the derivative of the activation function $\sigma$. For the first layer we compute

$$\partial_{x_i} z_k^0 = \sigma'(a_k^0) w_{ki}^0$$

We define the messages for the partial derivative as $m_{ki}^h := \partial_{x_i} z_k^h$. Therefore, we calculate the message passing as

$$m_{ki}^h = \sigma'(a_k^h) \sum_n w_{kn}^h m_{ni}^{h-1}$$

and

$$m_{ki}^0 = \sigma'(a_k^0) w_{ki}^0$$

Thus, the final equations for the backpropagation of the gradient are given by

$$
\boxed{
\begin{aligned}
& m_{ki}^0 = \sigma'(a_k^0) w_{ki}^0 \\
& \text{For } h = 1, \ldots, H-1 \\
& m_{ki}^h = \sigma'(a_k^h) \sum_n w_{kn}^h m_{ni}^{h-1} \\
& \partial_{x_i} V(\mathbf{x}) = \sum_n w_n^H m_{ni}^{H-1}.
\end{aligned}
}
$$

### A.4.2 The Hessian Computation

For the Hessian we compute

$$\partial_{x_j} \partial_{x_i} V(\mathbf{x}) = \partial_{x_j} \{ \sum_n w_n^H m_{ni}^{H-1} \}$$

$$= \sum_n w_n^H \partial_{x_j} m_{ni}^{H-1}.$$

The partial derivatives of the messages are

$$\partial_{x_j} m_{ki}^h = \sigma''(a_k^h) \left( \sum_n w_{kn}^h m_{ni}^{h-1} \right) \left( \sum_n w_{kn}^h m_{nj}^{h-1} \right) + \sigma'(a_k^h) \sum_n w_{kn}^h \partial_{x_j} m_{ni}^{h-1},$$

where $\sigma''$ denotes the second derivative of the activation function $\sigma$. For the first layer we compute

$$\partial_{x_j} m_{ki}^0 = \sigma''(a_k^0) w_{ki}^0 w_{kj}^0.$$

Next, we define the messages for the second order partial derivative as $\tilde{m}_{kij}^h := \partial_{x_j} m_{ki}^h = \partial_{x_j} \partial_{x_i} z_k^h$. Therefore, we calculate the message passing as

$$\tilde{m}_{kij}^h = \sigma''(a_k^h) \left( \sum_n w_{kn}^h m_{ni}^{h-1} \right) \left( \sum_n w_{kn}^h m_{nj}^{h-1} \right) + \sigma'(a_k^h) \sum_n w_{kn}^h \tilde{m}_{nij}^{h-1}$$

and

$$\tilde{m}_{kij}^0 = \sigma''(a_k^0) w_{ki}^0 w_{kj}^0.$$

Finally, the equations for the back-propagation of the Hessian are given by

$$
\tilde{m}^0_{kij} = \sigma''(a^0_k) w^0_{ki} w^0_{kj}
$$

For $h = 1, \ldots, H - 1$

$$
\tilde{m}^h_{kij} = \sigma''(a^h_k) \left( \sum_n w^h_{kn} m^{h-1}_{ni} \right) \left( \sum_n w^h_{kn} m^{h-1}_{nj} \right) + \sigma'(a^h_k) \sum_n w^h_{kn} \tilde{m}^{h-1}_{nij}
$$

$$
\partial_{x_j} \partial_{x_i} V(\mathbf{x}) = \sum_n w^H_n \tilde{m}^{H-1}_{nij}.
$$

# B  Algorithms

## B.1  The Thinning Algorithm

This section contains the thinning algorithm [27] to sample from the presented continuous-time POMDP model, see Algorithm 1.

---

**input** : $t$: time point
        $x$: state of the latent CTMC, i.e., $X(t) = x$
        $q_{\max} = \max_{u \in \mathcal{U}} \Lambda(x \mid u)$: maximum exit rate
**output** : $\xi$: Waiting time between the start time $t$ and the next jump of $X(t)$

Set current time to start time $T = t$
**while** *True* **do**
    Sample uniform variable for inverse CDF method $s \sim \mathrm{Uniform}(s \mid 0, 1)$
    Calculate minimum waiting time sample $\xi_{\min} = \frac{\log u}{q_{\max}}$
    Update current time $T = T + \xi_{\min}$
    Update the filtering dsitribution up to $T$, i.e., $\boldsymbol{\pi}_{[t,T)}$
    Draw $s' \sim \mathrm{Uniform}(s' \mid 0, 1)$
    **if** $s' \leq \Lambda(x \mid \mu(\boldsymbol{\pi}(T))$ **then**
        Calculate the waiting time $\xi = T - t$
        **return** *Waiting time $\xi$ and filtering distribution $\boldsymbol{\pi}_{[t,t+\xi)}$*
    **end**
**end**

---

**Algorithm 1:** Thinning algorithm

## B.2  The Collocation Algorithm

In this section we present the collocation algorithm used for the experiments. As a base distribution we use in the experiments e.g., a Dirichlet distribution $\boldsymbol{\pi}^{(i)} \sim \mathrm{Dir}(\boldsymbol{\pi} \mid \boldsymbol{\alpha})$. In Algorithm 2 the collocation algorithm can be found for the continuous discrete filter and a finite discrete observation space.

# C  Experimental Tasks

## C.1  Tiger Problem

In the tiger problem [9], the agent has to choose between two doors to open. Behind one door there is a dangerous tiger waiting to eat the agent, thus the agent is supposed to choose the other door to become free. Besides opening a door, the agent can also decide to wait and listen in order to localize the tiger. We adapted the problem to continuous time by defining the POMDP as follows: The state space $\mathcal{X}$ consists of the possible positions of the tiger, *tiger left* and *tiger right*. Executable actions of the agent are *listen*, *open left*, and *open right*. The tiger always stays at the same position therefore all the transition rates $\Lambda(x', x \mid u)$ are set to zero. When executing an *open* action, the agent receives reward with rate $0.1$ for the door without tiger and a negative reward of $-1.0$ for the door

**input** : $N$: Number of collocation samples

$p(\boldsymbol{\pi})$: Base distribution for collocation samples

$V_\phi(\boldsymbol{\pi})$: Function approximator for state value function with parameters $\phi$

$\bar{A}_\psi(\boldsymbol{\pi}, u)$: Function approximator for reparametrized advantage function with parameters $\psi$

**output** : $\{\hat{\phi}, \hat{\psi}\}$: parameters that have been fitted to approximately solve the HJB equation for $V_{\hat{\phi}}(\boldsymbol{\pi})$ and $\bar{A}_\psi(\boldsymbol{\pi}, u)$, respectively.

**for** $i = 1$ **to** $N$ **do**

    Sample collocation beliefs $\boldsymbol{\pi}^{(i)} \sim p(\boldsymbol{\pi})$

    **for** $u = 1$ **to** $N_u$ **do**

        Compute reset condition $\pi_+^{(i)}(x, y, u) = \frac{p(y|x,u)\pi^{(i)}(x)}{\sum_{x'} p(y|x',u)\pi^{(i)}(x')}, \forall x \in \mathcal{X}, y \in \mathcal{Y}$

        Compute Advantage values for the samples

$$A_\phi(i, u) = \sum_x R(x, u)\pi^{(i)}(x) + V_\phi(\boldsymbol{\pi}^{(i)}) + \tau \sum_{x,x'} \frac{\partial V_\phi(\boldsymbol{\pi}^{(i)})}{\partial \pi(x)} \Lambda(x', x \mid u)\pi^{(i)}(x')$$

$$+ \tau\lambda(\sum_x \pi^{(i)}(x) \sum_y V_\phi(\boldsymbol{\pi}_+^{(i)}(y, u)) - V_\phi(\boldsymbol{\pi}^{(i)})),$$

        where $\{\pi_+^{(i)}(x, y, u)\}_{x \in \mathcal{X}}$ are the components of $\boldsymbol{\pi}_+^{(i)}(y, u)$.

    **end**

    Compute best action $u^i = \arg\max_u A_\phi(i, u)$

**end**

Estimate parameters by solving

$$\hat{\phi} = \arg\min_\phi \sum_{i=1}^n (A_\phi(i, u^i))^2$$

Recompute all advantage values $A_{\hat{\phi}}(i, u)$ with fitted $\hat{\phi}$ and solve

$$\hat{\psi} = \arg\min_\psi \sum_{i=1}^n \sum_{u=1}^{N_u} (\bar{A}_\psi(\boldsymbol{\pi}^{(i)}, u) - \max_{u'} \bar{A}_\psi(\boldsymbol{\pi}^{(i)}, u') - A_{\hat{\phi}}(i, u))^2.$$

**Algorithm 2:** Collocation algorithm

with tiger. For executing the hearing action, the agent accumulates rewards of rate $-0.01$ but receives observations with a rate of 2, by hearing the tiger either on the left side (*hear tiger left*) or on the right side (*hear tiger right*). The received information is correct with probability 0.85 thus with probability 0.15 one hears the tiger at the opposite side. The discount factor $\tau$ is set to 0.9.

## C.2 Slotted Aloha Problem

The slotted aloha transition problem was introduced as POMDP in [8] and deals with decentralized control of stations transmitting packages in a single channel network. The task of the problem is to adjust the sending rate of the stations while the only information about the past transmission state is received at random time points. The state space consists of the number of stations that have a package ready for sending and the past transmission state as observation thus the state space is given by $\mathcal{X} = \{0, \ldots n\} \times \{idle, transmission, collision\}$, where $n = 9$ was chosen to limit the number of states to 30. For our contiuous-time POMDP model, we consider a continuous-time adaptation of [8]: Observations arrive with rate 0.5 and contain the past transmission state of the system which contained in the current state, thus $\mathcal{Y} = \{idle, transmission, collision\}$. While the maximum number of packages is not reached, new packages arrive with rate 0.5, leaving the past transmission state unchanged. The stations can send a package simultaneously with rate 5 but do not need to. The action $\rho$ represents the probability with which a package is actually send by a station, resulting in an actual send rate of $5 * \rho/n$.

Figure 4: Advantage updating method applied to slotted aloha problem.

The probability for transmission states given that $n$ packages are available are calculated as

$$p(idle \mid \rho) = (1 - \rho)^n$$
$$p(transmission \mid \rho) = \rho(1 - \rho)^{n-1}$$
$$p(collision \mid \rho) = 1 - p(idle \mid \rho) - p(transmission \mid \rho)$$

In case of perfect information, the optimal probability for transmission $\rho^*$ can be calculated by maximizing the probability of successful transmission which can be easily obtained by setting the derivative to zero resulting into

$$\rho^* = \arg\max_{\rho} \rho(1 - \rho)^{n-1} = \frac{1}{n}$$

This motivates the discretization of the actions resulting in $\mathcal{U} = \left\{\frac{1}{n}\right\}_{n=1,\dots,9}$.

The result of the advantage updating method, analogue to the one presented in the main text is depicted in Fig. 4.

### C.3 Gridworld Problem

We consider an agent moving in a $6 \times 6$ gridworld with a goal at position $(3, 2)$. There are four actions $\mathcal{U} = \{up, down, left, right\}$ indicating the direction the agent wants to move next. In our continuous-time setting, the agent moves at an exponentially distributed amount of time with rate of 10. With a probability of $0.7$, it moves into the indicated direction, but with probability $0.1$ moves into one of the other three directions due to slipping. Movement to invalid fields such as walls is not possible thus for those fields the transition probability is set to zero and the remaining probabilities are renormalized. Being at the goal position provides the agent with a reward with rate 1, otherwise no reward is accumulated. The agent receives noisy signal about his current position at a rate of 2. The signal indicates a field which sampled from a discretized 2D Gaussian distribution with standard deviation of $0.1$ centered at the position of the agent.

The result of the collocation method, analogue to the one presented in the main text is depicted in Fig. 5.

## D Hyper-Parameters

### D.1 Global Hyper-Parameters

Throughout the experiments, we use the following hyper-parameters:

Figure 5: Collocation method applied to gridworld problem.

- The neural networks are parametrized as:

```
linear1 = nn.Linear(in_dim, in_dim)
linear2 = nn.Linear(in_dim, in_dim)
linear3 = nn.Linear(in_dim, out_dim)

h1 = sigmoid(linear1(x))
h2 = sigmoid(linear2(h1))
out = linear3(h2)
```

- The advantage function network uses

$$\texttt{in\_dim} = |\mathcal{X}|$$
$$\texttt{out\_dim} = |\mathcal{U}|$$

- The value function network uses

$$\texttt{in\_dim} = |\mathcal{X}|$$
$$\texttt{out\_dim} = 1$$

- We use the Adam optimizer with learning rate $\alpha = 10^{-3}$.
- For training we use mini-batches of size $N_{\mathrm{b}} = 256$.

For the collocation method we use:

- $N_{\mathrm{o}} = 1$ optimization step for each mini-batch.
- A training for $N_e = 10000$ episodes.
- A discount decay by scheduling the decay parameter $\tau$ for the first $500$ steps of optimization, see e.g., [45].

The advantage updating method uses:

- $N_{\mathrm{s}} = 20$ optimization steps per episode, where a mini-batch is sampled from the replay buffer and one optimization step is carried out per mini batch.
- A training for $N_e = 1000$ episodes.
- An exploration process with a damping factor $\kappa = 7.5$ and a decaying noise variance $\sigma \in [1.5, .5]$. For the initial perturbation $\epsilon(u, t = 0)$, we sample the initial condition from the stationary distribution of the Ornstein-Uhlenbeck process.

## D.2 Experiment Dependent Hyper-Parameters

For the episode length in the advantage updating algorithm we use:

- Tiger Problem: $l_e = 10$.
- Slotted Aloha Problem: $l_e = 20$.
- Gridworld Problem: $l_e = 5$.

The initial belief for the problems is sampled from the following distributions:

- Tiger Problem: Uniform from the interval $[0.0, 1.0)$.
- Slotted Aloha Problem: Dirichlet distribution with concentration parameter $0.1$ for each state dimension.
- Gridworld Problem: Dirichlet distribution with concentration parameter $0.1$ for each state dimension.

We sub-sample the episodes for the advantage updating algorithm using the following number of samples:

- Tiger Problem: $N_s = 1000$.
- Slotted Aloha Problem: $N_s = 1000$.
- Gridworld Problem: $N_s = 100$.