[Reviews · NeurIPS 2020]

Review 1

Summary and Contributions: The manuscript presents an optimal control approach for a partially observable Markov decision process (POMDP) in continuous time with discrete state-spaces. The contribution is the formulation of a continous-time POMDP model and an algorithm to solve the Hamilton-Jacobi-Bellman equation using a neural network approximation to the values function

Strengths: The considered problem is - as far as I know - novel. The theoretical derivation are sound

Weaknesses: Post-rebuttal: I really appreciate the thoughtful response by the authors. The main issue for me was the motivation of this work and I would like to see good reasons that this new modelling paradigm is not only considered out of curiosity. I understand now, that it is an interesting problem to study but I am still unsure about the practical relevance since the control itself will always be implemented in some sort of (digital) time-discrete fashion. ======================================== I am not sure how relevant the particular problem formulation might be for the community, as a time discretization seems to be more convenient in many cases. The experimental evaluation is weak. In particular the gridworld problem is an example, which would have been much easier to solve in the continuous state space domain and it cannot be expected to scale well for larger dimensions.

Correctness: Yes, I couldn't find any flaws.

Clarity: Yes, I think for the available space, the authors have done their best to clearly formulate their contribution. However this topic requires background knowledge from different areas and is therefore not easy to follow.

Relation to Prior Work: The related work is properly reviewed. However, I would like to the in the experiment a comparison to alternative methods (even if they operate in discrete time or continuous state spaces)

Reproducibility: Yes

Additional Feedback:


Review 2

Summary and Contributions: This paper makes two contributions: (i) a theoretical contribution in which they extend the theory of POMDPs to continuous-time and finite discrete-state processes and (ii) algorithms for solving the analogue of the Bellman equations (which are called Hamilton-Jacobi-Bellman) together with a discussion of experimental results on toy examples. The theoretical contribution builds on work from the theory of stochastic filtering as well as standard material from RL. The algorithm is guided by ideas from the more traditional setting but essentially they have to solve a PDE with time derivatives so it is not entirely a routine matter to extend standard algorithms.

Strengths: The main strength is that introduces a whole new paradigm and develops the mathematical framework in a principled way. There is a lot of new machinery to digest but they do a good job in explaining the setup. There is too much to be really explained properly in detail but they do well with the limited space that they have.

Weaknesses: I think that the algorithmic work is very preliminary but it is fair enough for the first paper in this area. There is a vast amount of new material that comes with continuous time; they did a good job, but many parts of the framework could not be explained in the space they have. At some point they seem to veer towards continuous space and then shy away from it. I am a little puzzled that they keep writing the phrase "finite countable set". A finite set is *automatically* countable, a countable set may not be finite. More usually one sees the phrase "countably infinite" set but "countable finite" sounds silly and makes me have passing doubts about their familiarity with the subject. However, they have explained the harder mathematical ideas quite well.

Correctness: Yes, the theory seems correct as far as I could check. The empirical results are good but very preliminary.

Clarity: Yes and no. The material is promising and the explanations are as good as one expect in paper of this size. However, a serious reader would have to learn quite a lot of things in order to understand the points. The continuous-time world is very different from the 'real world"

Relation to Prior Work: Yes.

Reproducibility: Yes

Additional Feedback: Please explain what are the obstacles to having continuous time and space. The framework of HJB and trajectories will all work. One will have to introduce Feller-Dynkin semigroups or something like SDEs. ======================================= Post-rebuttal. I thank you for your well thought out rebuttal. I liked the paper before and I am still in favour of acceptance.


Review 3

Summary and Contributions: ***UPDATE: I have read the response. The core motivations are still unclear, and it is still unclear why discretization in time is a problem - I don't think you need pseudo-observations, I think perhaps you need to marginalize out [or leave as unobserved] time-bins without data. But I am not an expert. *** This paper presents a new model for decision making under uncertainty that involves discrete states / actions, but continuous time. The bulk of the paper is dedicated to a complex technical exposition that outlines the model, defines a notion of value function, and shows the existence of a Bellman-like recursion. Two approximate algorithms for learning the value function are given. Some small experiments illustrate them functioning in small domains.

Strengths: Overall, this is an interesting paper. It is technically sophisticated, and presents an interesting new model class. A few foundational quantities are derived that could be useful for future work. The learning algorithms seem straightforward. The experiments seemed reasonable, and illustrated the working of the algorithms.

Weaknesses: While I felt that this model class was interesting, I did not find it particularly well motivated. This is a complex model; why exactly do we need it? What advantages does it really have over some sort of discretization in time? In a related vein, it wasn't clear at all how the learning algorithms really worked. I am not an expert in this field, and I had a hard time following all of the proofs, so it wasn't clear to me exactly how these algorithms were approximating various intractable quantities. A few more high-level pointers would have been very helpful.

Correctness: Given the technical complexity of the work and my own lack of expertise, I cannot judge this well, although what I did understand seemed correct.

Clarity: It is generally well written, although complex.

Relation to Prior Work: Yes.

Reproducibility: No

Additional Feedback:


Review 4

Summary and Contributions: Post-rebuttal: I would like to thank the authors for the thoughtful response. The main issue for me was clarity, and I'm happy that the authors agreed to improve this aspect of the paper. However, it's hard to increase my score based on this promise alone. Nevertheless, my recommendation should really be considered a borderline recommendation. I will not fight against accepting this paper. ======================================== This paper addresses planning in settings with partial observability and a discrete state space, but with continuous time. This involves both filtering and control. To the best of my understanding, the contributions of this paper are: 1. Defining a model which captures such problems (a POMDP with continuous time), 2. Developing the Hamilton-Jacobi-Bellman (HJB) equation for the optimal value function in such a model, and 3. Approximating the optimal solution using both an offline and an online approach.

Strengths: The problem addressed is interesting and relevant. The proposed approach works well on the empirical evaluation.

Weaknesses: I found the paper hard to follow, and it wasn't even clear to me at first where the contributions of this paper start. The empirical evaluation is limited to very small toy problems. There is some related work using automata models to describe very similar problems, e.g.: * Lars Blackmore, Stanislav Funiak, Brian C. Williams, "Combining Stochastic and Greedy Search in Hybrid Estimation", AAAI 2005: 282-287 * F. Zhao, X. Koutsoukos, H. Haussecker, J. Reich, and P. Cheung, “Distributed monitoring of hybrid systems: A model-directed approach,” IJCAI 2001, pp. * S. Narasimhan and G. Biswas, “An approach to model-based diagnosis of hybrid systems”, in HSCC 2002, pp. 308–322

Correctness: As far as I could tell

Clarity: I found it hard to follow

Relation to Prior Work: Some related work is missing

Reproducibility: Yes

Additional Feedback:

[Author Response · NeurIPS 2020]

We would like to thank all reviewers for their constructive feedback and insightful comments. Here, our answers:

**Comments to all reviewers** First, we want to emphasize that our work is the first step towards the control of partial-observable continuous time discrete state space systems using modern machine learning methods. As the reviewers have clearly noticed and we also mentioned in the paper, the method in its current form is limited to low dimensional problems, since we solve the optimal non-linear filtering problem exactly. The toy problems function to explain the intuition behind the presented methodology and show that the proposed method works in principle. As the theory for the method in its current form is already quite complex, we think that further approximation methods would go way beyond the scope of a single paper. That being said, we will incorporate further explanations of the model and the underlying methodology to increase comprehensibility and to emphasize its applicability. This will also include discussing further related work areas such as stochastic hybrid systems and queueing networks, which are used, e.g., for TCP congestion control. We will add a short discussion on the control of partial observable continuous state space problems. The solution of these problems, however, involves even more difficulties since they require the stochastic optimal control framework under stochastic PDE dynamics and is thus left unsolved for future work.

**Reviewer #1** Time discretization is definitively possible but would go along with the need for pseudo observations at time points at which no observations are emitted by the latent process. This matter even represents a modelling problem on its own. Furthermore, the main benefit of not having an a priori time discretization of the problem is that the analysis can be carried out in continuous time as described in the paper. This way, the numerical errors occurring due to time discretization on a digital computer are controlled by sophisticated numerical differential equation solvers, which control the error automatically, by, e.g., adaptive step size methods.

The gridworld problem is one of the standard benchmarks for discrete spaces. From our point of view, a continuous space relaxation of this problem would defeat its purpose. Even fully-observable continuous state and continuous time problems are historically solved by a state space discretization, as in the work of [1]. Additionally, the partial observable case involves even more difficulties, for further details see Reviewer #2.

In our opinion, a direct comparison to advanced discrete time POMDP solvers would not be fair. Our proposed method does not aim at outperforming discrete methods for small problems but provides a principled way to approach continuous time discrete space problems. The main advantage to discrete methods becomes substantial when further approximations of the filtering distributions are introduced and larger problems are tackled. As already described, we leave the introduction of these approximations for future work to limit complexity.

**Reviewer #2** Sorry for the misnomer, we will scrap the word "countable" in "finite countable set". We just thought of the corresponding term for "countably infinite" which led to this accident.

As you already noted, continuous state space problems would involve, e.g., SDE dynamics of the latent process and the observation process. Therefore, the control of continuous state space problems under partial observability involves the control of the filtering distribution, which is described by a probability density. As the time evolution of the density is given by the Kushner-Stratonovich, a stochastic PDE, even simulating the system under a policy is a very hard problem. The main difficulty here is that the sufficient statistics of a general probability density are infinite dimensional. To approach this problem, one would have to resort to finding an efficient projection to a finite dimensional space by, e.g., looking at the first and second order moments. It is well known that for some special cases the filtering distribution has finite dimensional sufficient statistics. For example for a linear observation process with linear latent dynamics, the Kalman filter describes the evolution of the first and second order moments of the Gaussian filtering density. One then could use the finite dimensional mean and covariance inside the function approximator to estimate the value function even for arbitrary reward functions in contrast to the LQG method. However, for non-linear problems one would have to resort to approximations which find a finite dimensional projection, as it is done in, e.g., assumed density filtering.

**Reviewer #3** We will use the extra page of the camera-ready version for additional explanations to make the intuition behind the methodology clear and increase comprehensibility. Regarding your comment on time discretization, several new difficulties would have to be discussed as mentioned in Reviewer #1.

**Reviewer #4** The related work which you suggested, addresses the problem of time-discrete approximate optimal filtering in hybrid systems. In our paper, the equations for optimal filtering are by no means a contribution but originate from prior work which we referenced in the corresponding section. Instead, the focus lies on the problem of controlling these partial-observable systems. A further minor difference is that we approach the problem in continuous time as many original processes are likely to naturally evolve this way. However, the topic of approximate optimal filtering and hybrid systems seems a fruitful direction for future work when considering real world applications. We will therefore add relations to these topics as future work directions to the conclusion section.

**References**

[1] H. Kushner and P. G. Dupuis. *Numerical methods for stochastic control problems in continuous time*, volume 24. Springer Science & Business Media, 2013.


[Meta-Review · NeurIPS 2020]

The paper describes new offline and online techniques to optimize the policy of continuous time discrete state and action POMDPs. This paper makes an important contribution to the RL and control literature. Very little work has focused on continuous time control problems in the ML community. While the techniques assume that the model is known, do not scale to high dimensional problems and were tested only on toy problems, they introduce new formalisms that will help the community get familiar with the mathematics of continuous time control. Hence this paper will be of high interest for the RL community.